# GENERALIZING TO NEW DYNAMICAL SYSTEMS THROUGH FIRST-ORDER CONTEXT-BASED ADAPTATION

## ABSTRACT

In this paper, we propose FOCA (First-Order Context-based Adaptation), a learning framework to model sets of systems governed by common but unknown laws that differentiate themselves by their context. Inspired by classical modeling-and-identification approaches, FOCA learns to represent the common law through shared parameters and relies on online optimization to compute system-specific context. Due to the online optimization-based context inference, the training of FOCA involves a bi-level optimization problem. To train FOCA efficiently, we utilize an exponential moving average (EMA)-based method that allows for fast training using only first-order derivatives. We test FOCA on polynomial regression and time-series prediction tasks composed of three ODEs and one PDE, empirically finding it outperforms baselines.

## 1 INTRODUCTION

Scientists and engineers have made tremendous progress on modeling the behavior of natural and engineering systems and optimizing model parameters to best describe the target system (Ljung, 2010). This *modeling and system identification* paradigm has made remarkable advances in modern science and engineering (Schrödinger, 1926; Black & Scholes, 1973; Hawking, 1975). However, applying this paradigm to complex systems is difficult because the mathematical modeling of systems requires a considerable degree of domain expertise, and finding the best system parameters requires massive experimentation.

The availability of large datasets and advances in deep learning tools have made it possible to model a target system without specific mathematical models, relying instead on flexible model classes (Brunton et al., 2016; Gupta et al., 2020; Menda et al., 2020; Jumper et al., 2021; Kochkov et al., 2021; Degrave et al., 2022). However, when the characteristics of target systems change (e.g., system parameters, boundary conditions), the flexibility of data-driven models makes them difficult to adapt.

Deep learning approaches typically handle the contextual change by collecting data from the new behavioral mode and re-training the model on the new dataset. However, this approach can be impractical, especially when the system is complex and context change is frequent. We are interested in developing a framework that learns a common shared model of the systems and inferring the context that best describes the target system to predict response. Our study considers a target system whose input $x$ and response $y$ can be described by $y = f(x, c)$, where $f$ denotes the function class shared by the target systems and $c$ denotes the system-specific context.

One possible approach for modeling such target systems is meta-learning (Hospedales et al., 2021), which learns how to adapt to new systems. Meta-learning is typically a combination of an adaptive mechanism and training for the adaptation. One typical meta-learning approach is to use an encoder that takes the adaptation data and returns the learned context (Santoro et al., 2016; Mishra et al., 2018; Garnelo et al., 2018; Kim et al., 2019). Although encoder-based adaptation schemes require constant memory usage, their parameterized encoders limit the adaptation capability. Other approaches (pre) train the parameters on the dataset collected from the various modes and update all parameters using gradient-descent (Finn et al., 2017; Nagabandi et al., 2018; Rajeswaran et al., 2019). Despite their effective adaptability, those approaches are often prone to (meta) over-fitting (Antoniou et al., 2019), especially when the adaptation target is complex and adaptation data is scarce. Instead of updating all parameters, Raghu et al. (2019); Zintgraf et al. (2019) update a subset

Table 1: Comparison of context-based generalization approaches. The memory column specifies the additional memory requirements for adaptation during the training phase of the algorithms. $|\cdot|$ denotes the number of elements of $\cdot$ .

| | Training models (parameters) | Adaptation mechanism | Memory |
|---|---|---|---|
| Encoder (Mishra et al., 2018; Lee et al., 2020a) | $f_\theta, g_\phi$ | $\hat{c} = g_\phi(\mathcal{D})$ | $\mathcal{O}(|\phi|)$ |
| CAVIA (Zintgraf et al., 2019) | $f_\theta$ | $\hat{c}_{k+1} = \hat{c}_k - \lambda\nabla_{\hat{c}_k}\mathcal{L}\sum_{(x,y)\in\mathcal{D}}(f_\theta(x,\hat{c}_k),y),$ $\hat{c}_0 = 0, \hat{c} = c_K$, where $K$ is the adaptation steps. | $\mathcal{O}(|\theta| \cdot K)$ |
| CoDA (Kirchmeyer et al., 2022) | $f_\theta, \boldsymbol{W}$ | $\hat{c} = \arg\min_c \sum_{(x,y)\in\mathcal{D}}\mathcal{L}(f_{\theta+\boldsymbol{W}c}(x),y)$ | $\mathcal{O}(|\boldsymbol{W}|)$ |
| FOCA | $f_\theta$ | $\hat{c} = \arg\min_c \sum_{(x,y)\in\mathcal{D}}\mathcal{L}(f_{\bar{\theta}}(x,c),y),$ where $f_{\bar{\theta}}$ is an EMA copy of $f_\theta$. | $\mathcal{O}(1)$ |

of parameters, which we call context $\hat{c}$, while fixing the remainder. Although this modeling approach is effective, the training requires computing higher-order derivatives.

Many training method for meta-learning have been proposed (Finn et al., 2017; Nichol et al., 2018; Rajeswaran et al., 2019; Deleu et al., 2021). Typically encoder-based meta-learning trains an encoder and a prediction model jointly and no online optimization-based adaptation occurs. The training of gradient-based meta-learning is typically cast as bi-level optimization. Typical gradient-based meta-learning is often carried out by propagating through the update steps, which requires higher-order derivative calculations. To avoid such issues, Finn et al. (2017); Nichol et al. (2018) propose a first-order approximation of derivatives to update the (meta) parameters. Alternatively, the implicit gradient method can be used to lower the computational burden (Rajeswaran et al., 2019), but gradient computation errors can be significant and result in performance degradation (Liao et al., 2018; Zhou et al., 2019; Blondel et al., 2021; Chen et al., 2022).

In this paper, we propose FOCA (First-Order Context-based Adaptation), a context-based meta-learning method that is specialized for complex dynamical systems whose behavior can be characterized by a common mathematical model $f$ and a context $c$. Specifically, FOCA considers target systems of the form $y = f_\theta(x,\hat{c})$, where $f_\theta$ denotes the learned function class shared by the target systems and $\hat{c}$ denotes the inferred system-specific context. FOCA learns the function class $f_\theta$ during training and solves the system identification problem through numerical optimization to find the proper $\hat{c}$. The online context optimization bypasses the limitation of encoder-based approaches, but it entails a higher computational burden. To train FOCA efficiently, we thus propose using an exponential moving average (EMA) based training method, which operates with first-order derivatives and no additional memory usage. From our experiments, we also confirm that EMA-based training decreases the computational burden of training and improves the generalization performance of FOCA. The contributions of this work are summarized as follows:

- We propose FOCA, a learning framework specialized for modeling complex dynamical system families that are described by an unknown common law and system-specific context.

- We propose an EMA-based training method for training FOCA that overcomes the burden of second-order derivative calculations while showing better generalization results compared to other training methods.

- We empirically demonstrate that FOCA outperforms or is competitive to various meta-learning baselines in static function regression and time-series prediction tasks evaluated both in-distribution and out-of-distribution.

## 2 RELATED WORK

**Learning to generalize to the new systems.** Transfer learning (Zhuang et al., 2020) attempts to generalize learned models to new systems (or tasks) by fine-tuning a small number of parameters to the new tasks. As a more direct way of generalization to new systems, meta-learning learns how to adapt quickly to new systems. In particular, gradient-based meta-learning (GBML) approaches perform few-step gradient descent updates of the model parameter $\theta$ for adaptation (Finn et al., 2017; Nichol et al., 2018). Focusing on the empirical evidence that the plain GBML is prone to overfit the training tasks, CAVIA (Zintgraf et al., 2019) performs gradient-based updates for a small

dimensional context vector $c$. On the other hand, encoder-based meta-learning methods jointly train an encoder network $g_\phi$ alongside the prediction model $f_\theta$ (Santoro et al., 2016; Mishra et al., 2018; Garnelo et al., 2018; Kim et al., 2019; Gordon et al., 2020).

A few works investigate how to solve our target problem (i.e., learning a common unknown rule $f$ of static or dynamic systems with the learned adaptation mechanism). Lee et al. (2020b) applies GBML to generalize over Hamiltonian systems. LEADS (Yin et al., 2021) jointly learns the system specific networks $g_\phi$ and shared network $f_\theta$, and predicts the response of the dynamic system with $f_\theta + g_\phi$. CoDA (Kirchmeyer et al., 2022) learns the common model parameters $\theta$, a linear basis $\boldsymbol{W}$, and context vectors $\hat{c}$ while defining $f_\theta(\cdot, \hat{c})$ as $f_{\theta + \boldsymbol{W}\hat{c}}(\cdot)$. To adapt to the new system with the observation dataset $\mathcal{D}$, it solves an optimization, $\min_c \sum_{(x,y)\in\mathcal{D}} \mathcal{L}\left(f_{\theta+\boldsymbol{W}c}(x), y\right)$. In contrast, FOCA adapts by solving Eq. (6), which is an effective method to generalize to systems that do not require any structural assumptions. This approach is different from LEADS, where the adaptation is sum-decomposable, and from CoDA, where the adaptive parameters are a span of a linear basis. We summarize the differences between FOCA and baselines in Table 1.

**Solving bi-level optimization to learn.**    Training FOCA involves solving a bi-level optimization. Differentiable optimization layers (Amos & Kolter, 2017; Agrawal et al., 2019; Meng et al., 2021) encapsulate an optimization problem as a layer of neural networks. As they employ optimization as a layer of the network, the training becomes a bi-level optimization problem. The backward path of differentiable optimization layers is often implemented by backpropagating the optimization steps or implicit gradient methods. As a special case, Bertinetto et al. (2019) employs inner optimization problems that can be solved analytically.

In the context of meta-learning, Bertinetto et al. (2019) considers solving the analytically solvable inner optimization problem. However, such approaches limit the flexibility of network parameterization or adaptation mechanisms. As a more flexible approach, Implicit MAML (Rajeswaran et al., 2019) extends MAML to overcome the extensive memory usage of adaptation and allows for more inner adaptation steps without memory issues. However, implicit gradient methods can calculate inexact meta-gradients (Deleu et al., 2021), and, as we show in the experiments, such errors can degrade performance. Instead, we propose to solve the bi-level optimization training problem with the EMA method, which also has constant memory usage. We empirically show that the EMA-based training method outperforms implicit gradient-based methods on test metrics.

## 3    PRELIMINARIES

This section reviews classical parameter identification and meta-learning as preliminaries for explaining the proposed method.

**System identification.**    Consider the response $y$ from an input $x$ of a system $f$ with an unknown (or unobservable) context $c$,

$$y = f(x, c). \tag{1}$$

Classical parameter identification aims at inferring the (optimal) context $c^*$ with data from a task $\mathcal{D} = \{(x_n, y_n)\}_{n=1}^N$. Under the assumption that $(x_n, y_n)$ pairs in $\mathcal{D}$ are generated from the same system, the identification process can be formulated as an optimization problem as follows:

$$c^* = \arg\min \sum_{n=1}^N \mathcal{L}(f(x_n, c), y_n), \tag{2}$$

where $\mathcal{L}$ is a discrepancy metric. By using the optimized context $c^*$, we can query responses of the (identified) system $f(\cdot, c^*)$ to inputs.

**Meta-learning.**    Meta-learning is a collection of algorithms that learn to adapt quickly to a new task by learning from experiences of related tasks. Assuming that tasks (i.e., the systems of different contexts) are drawn from a distribution $p(\mathcal{T})$ and each task includes inference data $\mathcal{D}_{\text{inf}}^{\mathcal{T}}$ and

**Training**

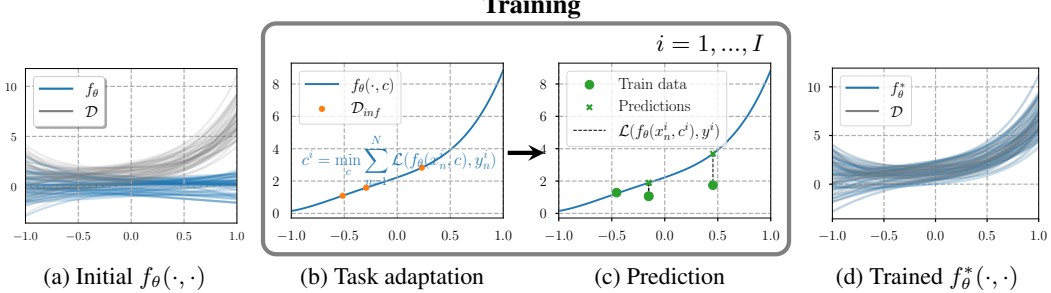

(a) Initial $f_\theta(\cdot, \cdot)$      (b) Task adaptation      (c) Prediction      (d) Trained $f_\theta^*(\cdot, \cdot)$

Figure 1: Training of FOCA: (a) the function class captured by the initial $f_\theta(\cdot, \cdot)$ and task distribution $\rho(\tau)$, (b) inference of $c^i$ by solving Eq. (6) with context inference data and identification of $f_\theta(\cdot, c^i)$, (c) prediction of $f_\theta(\cdot, c^i)$ and the outer loss $\mathcal{L}(f_\theta(x_n^i, c^i), y_n^i)$, (d) the optimized $f_\theta^*(\cdot, \cdot)$. FOCA is trained by repeating (b), (c), and minimizing $\mathcal{L}(f_\theta(x_n^i, c^i), y_n^i)$.

evaluation data $\mathcal{D}_{\text{ev}}^{\mathcal{T}}$, the meta-learning objective can be defined as follows:

$$\min_{\theta, \psi} \mathbb{E}_{\mathcal{T} \sim p(\mathcal{T})} \left[ \mathcal{L} \left( \mathcal{D}_{\text{ev}}^{\mathcal{T}}, f_{\theta'} \right) \right] \tag{3}$$

$$\text{s.t.} \quad \theta' = g_\psi \left( \mathcal{D}_{\text{inf}}^{\mathcal{T}}, \theta \right), \tag{4}$$

where $g_\psi(\cdot, \theta)$ is a (meta-)learned adaptation algorithm that is parameterized by $\theta$ and $\psi$, $f_{\theta'}$ is the task-specific model that is parameterized by the adapted parameter $\theta'$, and $\mathcal{L}$ is a loss metric. In summary, meta-learning learns $\theta$ and $\psi$ to infer a model from $\mathcal{D}_{\text{inf}}^{\mathcal{T}}$ that minimizes the error on $\mathcal{D}_{\text{ev}}^{\mathcal{T}}$. Optimized meta-parameters $\theta^*$ and $\psi^*$ can then be used to "quickly" adapt to new tasks from $p(\mathcal{T})$.

## 4 METHODOLOGY

In this section, we present First-Order Context-based Adaptation (FOCA) to learn to jointly identify and predict systems in the same family. We first introduce the training problem of FOCA and then discuss the EMA-based training method for efficiently solving the training problem. We also provide our code as supplementary material.

### 4.1 PROBLEM FORMULATION

We formulate the learning problem of FOCA as a bi-level optimization where the inner optimization Eq. (6) is for inferring the context, and the outer optimization Eq. (5) is for learning $f_\theta$ conditioned on the identified context. The proposed optimization problem is as follows:

$$\min_\theta \sum_{\mathcal{D}^i \in \mathcal{D}^{\text{tr}}} \sum_{(x_n^i, y_n^i) \in \mathcal{D}_{\text{ev}}^i} \mathcal{L} \left( f_\theta(x_n^i, \hat{c}^i), y_n^i \right) \tag{5}$$

$$\text{subject to} \quad \hat{c}^i = \arg\min_c \sum_{(x_n^i, y_n^i) \in \mathcal{D}_{\text{inf}}^i} \mathcal{L} \left( f_\theta(x_n^i, c), y_n^i \right), \tag{6}$$

where $\mathcal{D}_{\text{inf}}^i$ and $\mathcal{D}_{\text{ev}}^i$ are the context inference and evaluation datasets of the $i$th system, which satisfy $\mathcal{D}_{\text{inf}}^i \cap \mathcal{D}_{\text{ev}}^i = \emptyset$ and $\mathcal{D}_{\text{inf}}^i \cup \mathcal{D}_{\text{ev}}^i \subset \mathcal{D}^i$, $\mathcal{D}^{\text{tr}} = \{\mathcal{D}^i\}_{i=1,\dots}$ is a collection of meta-training datasets from multiple systems, $f_\theta$ is a prediction model parameterized by $\theta$, and $\mathcal{L}$ is a loss metric.

The prediction model $f_\theta$ is trained by solving Eqs. (5) and (6). The trained $f_\theta^*$ describes the common structure of the systems in $\mathcal{D}$. Identification of a specific function (i.e., adaptation) is done by solving Eq. (6) with the data points sampled from the specific system. After finding $c^i$, we can query for meta-test predictions with $f_\theta(\cdot, c^i)$ as visualized in Fig. 1 (b) and (c).

FOCA can be viewed as a special case of the meta-learning recipe explained in Section 3 by defining $\theta' = [\theta, \hat{c}^i]$ and setting Eq. (4) as Eq. (6). Even though FOCA is formulated similarly to optimization-based meta-learning methods, the optimization variables for the adaptation is $\hat{c}^i$ rather than $\theta$. In terms of implementation, this algorithmic selection allows us to batch-solve inner-level optimization with standard automatic differentiation tools. FOCA can therefore be trained efficiently. We can also interpret this clear separation of $\theta$ and $c$ as capturing function classes and their coefficients separately. We further inspect this view of FOCA in Section 5.1.

---

**Algorithm 1:** Training FOCA with exponential moving average (EMA)

---

**Input:** Prediction model $f_\theta$, training tasks $\mathcal{D}^{\text{tr}}$, inner optimization steps $K$, inner optimization
      step size $\alpha$, weighting factor $\tau$, context regularization weight $\lambda$

1   $\bar{\theta} \leftarrow \theta, f_{\bar{\theta}} \leftarrow f_\theta$                                      `// Initialize the target model`

2   **for** $1, 2, ...$ **do**

3      Sample batch of tasks $\mathcal{D}^{\text{b}} \sim \mathcal{D}^{\text{tr}}$

4      **for** $\mathcal{D}^i \in \mathcal{D}^b$ **do**

5         construct $\mathcal{D}_{\text{inf}}^i$ and $\mathcal{D}_{\text{ev}}^i$ from $\mathcal{D}^i$

6         $\hat{c}^i \leftarrow 0$

7         **for** $k = 1, ..., K$ **do**

8            $\mathcal{L}(c^i) = \sum_{(x_n^i, y_n^i) \in \mathcal{D}_{\text{inf}}^i} \mathcal{L}(f_{\bar{\theta}}(x_n^i, c^i), y_n^i) + \lambda \|c^i\|$

9            $\hat{c}^i \leftarrow \hat{c}^i - \alpha \nabla_{c^i} \mathcal{L}(c^i)$                        `// Solve Eq. (6)`

10        **end**

11     **end**

12     Evaluate $\mathcal{L}(\theta) = \sum_{\mathcal{D}^i} \sum_{(x_n^i, y_n^i) \in \mathcal{D}_{\text{ev}}^i} \mathcal{L}(f_\theta(x_n^i, \hat{c}^i), y_n^i)$

13     $\theta \leftarrow \theta - \alpha \nabla_\theta \mathcal{L}(\theta)$

14     $\bar{\theta} \leftarrow \tau\theta + (1 - \tau)\bar{\theta}$                               `// Update the target model`

15 **end**

---

## 4.2 TRAINING FOCA

Training FOCA with gradient descent involves computation of the partial derivatives of $\mathcal{L}^{\text{upper}} = \sum_{\mathcal{D}^i \in \mathcal{D}^{\text{tr}}} \sum_{(x_n^i, y_n^i) \in \mathcal{D}_{\text{ev}}^i} \mathcal{L}\left(f_\theta(x_n^i, \hat{c}^i), y_n^i\right)$ with respect to $\theta$. We express $\partial \mathcal{L}^{\text{upper}} / \partial\theta$ using the chain rule taking $\hat{c}^i$ as the intermediate variable,

$$\frac{\partial \mathcal{L}^{\text{upper}}}{\partial \theta} = \frac{\partial \mathcal{L}^{\text{upper}}}{\partial \hat{c}^i} \frac{\partial \hat{c}^i}{\partial \theta}. \tag{7}$$

Here, $\partial \mathcal{L}^{\text{upper}} / \partial \hat{c}^i$ can be computed via an automatic differentiation package. However, the computation of $\partial \hat{c}^i / \partial \theta$ is less straightforward because of the implicit relations between $\hat{c}^i$ and $\theta$. That is, their relations are defined through an optimization problem rather than a computational chain.

One possible approach to calculate $\partial \hat{c}^i / \partial \theta$ is to backpropagate through the inner optimization steps, which calculates the derivatives by unrolling the inner optimization steps through the computational graph. However, this method is not memory efficient, especially with a large number of model parameters $|\theta|$ or with many inner optimization steps. Another method is to approximate $\partial \hat{c}^i / \partial \theta$ with first-order methods as similarly done in Finn et al. (2017); Nichol et al. (2018). However, as reported by others (Zhou et al., 2019; Jayathilaka, 2019; Chen et al., 2022), this approach often requires techniques to mitigate the approximation errors. Another method calculates $\partial \hat{c}^i / \partial \theta$ using implicit gradients and can bypass the memory usage of the first approach. However, calculating the gradient without error requires solving the inner optimization problem optimally (Liao et al., 2018; Blondel et al., 2021). In practice, this restricts the expressivity of $f_\theta$ to architectures that can be solved optimally Eq. (6). We propose an efficient approach to train FOCA that uses no additional memory and allows for flexible architecture of $f_\theta$.

**EMA-based training method.** Inspired by recent developments in self-supervised learning (Grill et al., 2020; Caron et al., 2021) and the target-network of reinforcement learning (Lillicrap et al., 2016), we use a simple method based on the exponential moving average (EMA) to train FOCA. The core idea is to consider $\hat{c}$ as an independent input given from outside of the gradient chain. First, we copy $\theta$ to create a delayed target model $f_{\bar{\theta}}$ parameterized by $\bar{\theta}$. We use $f_{\bar{\theta}}$ to infer $\hat{c}^i$ with gradient descent on Eq. (6). We then update $\theta$ by using $\nabla_\theta \mathcal{L}(\theta) = \partial \mathcal{L}^{\text{upper}} / \partial \theta$ evaluated on $\hat{c}^i$. As the parameters of $f_\theta$ and $f_{\bar{\theta}}$ are separate, this avoids differentiating the argmin operator to calculate $\nabla_\theta \mathcal{L}(\theta)$. After updating $\theta$, we update $\bar{\theta}$ as $\bar{\theta} \leftarrow \tau\theta + (1 - \tau)\bar{\theta}$ with $0 \leq \tau \ll 1$. The training procedure is summarized in Algorithm 1.

The proposed training algorithm utilizes $f_{\bar{\theta}}$ (i.e., the delayed copy of $f_\theta$) to infer $\hat{c}^i$ by numerically solving Eq. (6). When we train $f_\theta$, it takes $\hat{c}^i$ as an input that contains meaningful information that

Figure 2: Polynomial regression results. All models infer the polynomial from the given context points (●). The prediction results and ground truth values are visualized with the blue-solid (—) and gray-dashed (- ·) lines, respectively.

differentiates the system $i$ from the other systems. At the early training phase, $\hat{c}^i$ might have no meaning (i.e., the different $\hat{c}^i$s do not change the prediction results). However, this phenomenon is resolved during training because $f_\theta$ is trained as if $\hat{c}^i$ is a (learned) system-differentiating context. Furthermore, disentangling the model for prediction and inference results in a practical advantage: we can employ any optimization algorithm (even non-differentiable ones) to solve Eq. (6). The proposed method shows its advantages in memory usage and implementation.

We also observed that EMA-based training can increase the generalization performance over the other training methods. We discuss this aspect with the experimental results in Section 5.3.

## 5 EXPERIMENTS

In this section, we discuss experiments that show the properties of FOCA with static regression problems (Section 5.1), demonstrate its effectiveness in predicting the response of dynamics systems (Section 5.2), and investigate the proposed EMA-based training method (Section 5.3).

**Baselines.** We compare FOCA to three baselines that adapt to the tasks by inferring $\hat{c}$. The first two baselines are representative methods that find $\hat{c}^i$ via the inference schemes discussed in Section 1. The last baseline, CoDA (Kirchmeyer et al., 2022), is our closest baseline in terms of application: a meta-learning method tailored for generalization of physical systems. To make fair comparisons, all models use the same network architecture $f_\theta(\cdot)$ for each task. The baselines are as follows:

- Encoder employs an encoder network to extract the context $\hat{c}^i$ from $\mathcal{D}_{\text{inf}}^i$. That is, $\hat{c}^i = g_\phi(\mathcal{D}_{\text{inf}}^i)$, where $g_\phi$ is the encoder network. We employ SNAIL (Mishra et al., 2018) and CaDM (Lee et al., 2020a)-based $g_\phi$ for static and time-series tasks, respectively.
- CAVIA (Zintgraf et al., 2019) performs a few gradient steps on Eq. (6) to extract the context $\hat{c}^i$ from $\mathcal{D}_{\text{inf}}^i$ and train $f_\theta$ via the backpropagation-through-optimization-step scheme.
- CoDA (Kirchmeyer et al., 2022) learns the common model parameters $\theta$, a linear basis $\mathbf{W}$, and context vectors $\hat{c}^i$ while defining $f_\theta(\cdot, \hat{c}^i)$ as $f_{\theta + \mathbf{W}\hat{c}^i}(\cdot)$. For adaptation to a new system, it solves the optimization problem $\min_c \sum_{(x_n^i, y_n^i) \in \mathcal{D}_{\text{inf}}^i} \mathcal{L}\left(f_{\theta + \mathbf{W}c^i}(x_n^i), y_n^i\right)$.

### 5.1 STATIC REGRESSION

We first examine polynomial function regression to understand the properties of FOCA. In this task, the objective is to identify a polynomial and make predictions from the identified functions as shown in Fig. 2. We prepare 4th order polynomial datasets by sampling the coefficients from $\mathcal{U}(0.1, 2.5)$, and set $|\mathcal{D}_{\text{inf}}^i|$ and $|\mathcal{D}_{\text{ev}}^i|$ as 5 and 15, respectively. Fig. 3 shows the average of the test root mean squared error (RMSE) values of models. As shown in Fig. 3, FOCA adapts to the different polynomials better than the baseline models. We provide the details of the model architectures and training scheme in Appendix A. We then investigate the properties of FOCA by using the polynomial dataset.

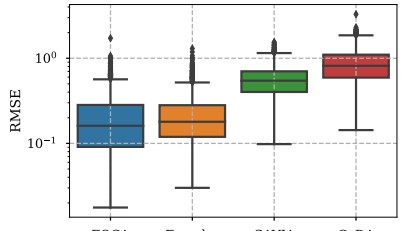

Figure 3: Static regression test RMSE.

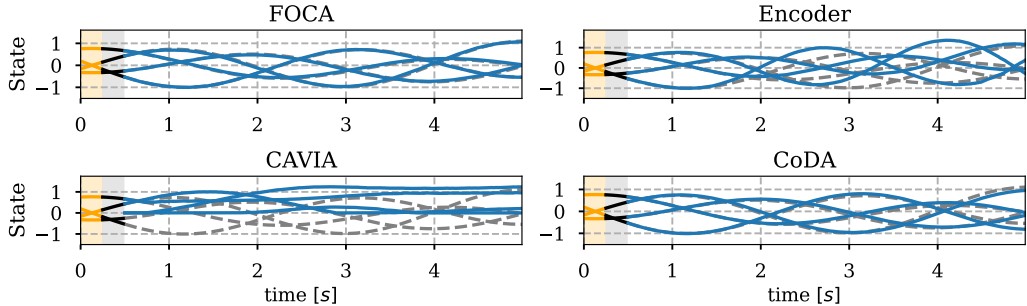

Figure 5: MS prediction results. All models infer the physical constants from historical (shaded in yellow) and current (shaded in gray) state observations, and then predict rollouts of future state trajectories (colored in blue). The gray dashed lines visualize prediction targets. FOCA successfully identifies the target systems and predicts long-term futures.

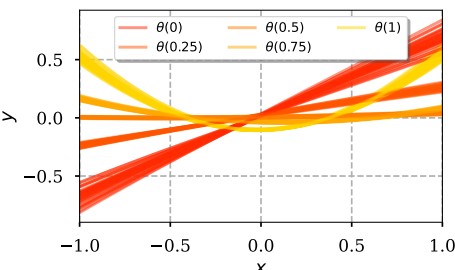

Figure 4: Role of $\hat{c}$ within model classes $\theta(\lambda)$ that interpolate between quadratic and linear.

**Do $\theta$ and $c$ contain function class and parameter information, respectively?** The roles of $\theta$ and $c$ are to capture a function class and to identify a specific function from that class, respectively. For example, $\theta$ should express the fact that functions are polynomials and $c$ should capture the coefficients of a specific polynomial. To analyze whether $\theta$ and $c$ fulfill these roles, we train two FOCA models: one with parameters $\theta_1$ meta-trained on linear functions, and one with parameters $\theta_2$ meta-trained on quadratic functions. We then define models that interpolate between these two sets of parameters with $\theta(\lambda) = (1 - \lambda)\theta_1 + \lambda\theta_2$ with $0 \leq \lambda \leq 1$. Fig. 4 shows predictions from $f_{\theta(\lambda)}(\cdot, c)$ with different samples of $c \sim \mathcal{U}(-0.025, 0.025)^{32}$ and different values of $\lambda$. We conjecture that $f_{\theta(\lambda)}(\cdot, c)$ represents linear functions when $\lambda = 0$, quadratic functions when $\lambda = 1$, and functions that interpolate between linear and quadratic functions when $0 < \lambda < 1$. We verify the conjectures by observing that within a single function class $\theta(\lambda)$, each $c$ parameterizes a different function from that class (e.g., the red curves are all linear but each of them is a different linear function).

## 5.2 TIME-SERIES PREDICTION

We evaluate FOCA on time-series prediction in three ODE datasets, including mass-spring systems (MS), Lotka-Volterra equations (LV), glycolytic oscillators (GO), and one PDE dataset derived from the Navier-Stokes equations (NS). In this setting, the objective is first to identify target systems from historical and current state observations and then to predict rollouts of future state trajectories as shown in Fig. 5. We provide details for dataset generation, model architectures, and training schemes in Appendices B to E.

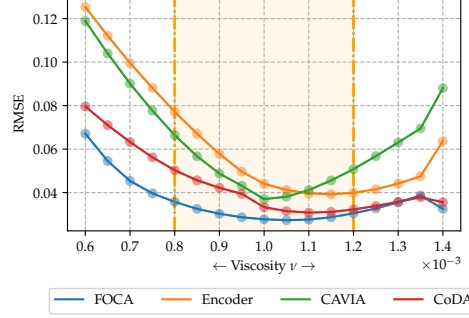

Figure 6: RMSE for different models on the Navier-Stokes equations at different viscosities $\nu$. The orange area denotes the training region.

**Task generalization.** We study the generalization performances of the models on systems whose context parameters are generated from inside as well as outside the training distribution (i.e., in-training and out-of-training distribution adaptation tasks). We evaluate the models by measuring the average and standard deviation of the rollout prediction root mean squared errors (RMSE) over the test systems. As shown in Table 2, FOCA generally outperforms the baseline algorithms, indicating that FOCA adapts well to new systems regardless of the parameters of the system. To further analyze the performance variation of models to the change of system parameters, we visualize the prediction

Table 2: In/out-of training distribution generalization results. Smaller is better ($\downarrow$). We measure the RMSE values per trajectory. Average RMSEs and their standard deviations are reported. Best in **bold**; second underlined.

| | MS | | LV | | GO | | NS | |
|---|---|---|---|---|---|---|---|---|
| | In-training | Out-of training | In-training | Out-of training | In-training | Out-of training | In-training | Out-of training |
| Encoder | $2.272 \pm 1.442$ | $2.379 \pm 1.506$ | $\underline{0.137} \pm 0.093$ | $0.326 \pm 0.362$ | $0.062 \pm 0.092$ | $\underline{0.156} \pm 0.174$ | $0.057 \pm 0.016$ | $0.114 \pm 0.026$ |
| CAVIA | $1.743 \pm 0.595$ | $1.793 \pm 0.628$ | $0.312 \pm 0.316$ | $0.706 \pm 0.551$ | $\underline{0.046} \pm 0.061$ | $0.157 \pm 0.161$ | $0.057 \pm 0.011$ | $0.114 \pm 0.028$ |
| CoDA | $\textbf{0.250} \pm 0.201$ | $\underline{0.545} \pm 0.402$ | $0.258 \pm 0.199$ | $0.642 \pm 0.349$ | $0.072 \pm 0.115$ | $0.269 \pm 0.258$ | $\underline{0.051} \pm 0.034$ | $\underline{0.095} \pm 0.046$ |
| FOCA | $\underline{0.258} \pm 0.156$ | $\textbf{0.478} \pm 0.295$ | $\textbf{0.079} \pm 0.061$ | $\textbf{0.303} \pm 0.291$ | $\textbf{0.043} \pm 0.066$ | $\textbf{0.147} \pm 0.168$ | $\textbf{0.042} \pm 0.026$ | $\textbf{0.070} \pm 0.038$ |

Figure 7: Target solutions and RMSE propagation on the Navier-Stokes equations for different models. FOCA manages to better contain errors even after long autoregressive rollouts and out-of-distribution context samples.

errors of the models on the Navier-Stokes equations for different viscosity values in Fig. 6, while in Fig. 7 we show a sample of target predictions and error propagation for out-of-distribution viscosity $\nu = 10^{-4}$. FOCA shows better prediction results than baselines, not only for new systems with in-distribution contexts but also for out-of-distribution ones. We provide the additional studies for MS, LV, and GO in Appendices B to D, including out-of-distribution error comparison visualizations.

## 5.3 TRAINING ABLATION STUDY

In this section, we perform ablation studies to understand the effect of the proposed training scheme on the performance of FOCA.

**Is the EMA target network scheme effective?** To train FOCA, we employ the delayed target network $f_{\hat{\theta}}$ to infer $\hat{c}^i$, which we consider an independent input when we train $f_\theta$ (EMA). However, we can consider employing other training methods as mentioned in Section 4.2. We compare against training FOCA by calculating gradients by backpropagating through the unrolled inner optimization steps (BPTO), approximating the inner loop gradient with a first-order approximation (FO), or employing the implicit method (e.g., Amos & Kolter 2017) for calculating the gradient of the inner optimization (Implicit). We compare the test set performance of each method on the LV dataset while applying the same hyperparameters. Fig. 8a visualizes the test performances of each method over the training steps.

As shown in Fig. 8a, EMA has the lowest test set errors. We conjecture that EMA outperforms Implicit because the implicit gradient estimator can have error in practice, as the inner optimization problem cannot be solved optimally, especially when $f_\theta$ is parameterized with a general neural network (Blondel et al., 2021; Liao et al., 2018). This issue might be resolved by employing a special structure to $f_\theta$ such as linearity or convexity. However, such treatment can prevent the use of arbitrary network architecture as $f_\theta$.

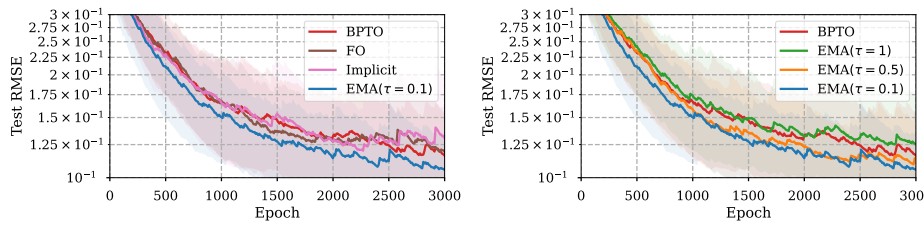

(a) Training method ablation results    (b) $\tau$ ablation results

Figure 8: Training method ablation results. We train each method five times with different random seeds. The bold lines and shadow region illustrate the averages and standard deviations of the training runs. We smooth the curves using a moving average.

We also note that `EMA` outperforms `BPTO`. Because `BPTO` calculates the gradient by chained computation on the computational graph (at the cost of memory usage), it is difficult to claim that the performance degradation is related to the gradient computation. We conjecture this phenomenon is related to the consistency of $\hat{c}$ estimation. It is noteworthy that `BPTO` is equivalent to CAVIA with a larger inner gradient step. Hence, FOCA is also superior to CAVIA with the same number of inner optimization steps. The results indicate that `EMA` trains FOCA effectively while allowing us to employ arbitrary network architecture and lower memory usage.

**How does $\tau$ affect the performance?**  We investigate the effect of the delay parameter $\tau$ in our training scheme. Our training scheme assumes $\hat{c}$ contains enough information to differentiate a specific system. As it is considered an input, the consistency of $\hat{c}$ over the training step becomes crucial to the stable training of $f_\theta$. In this sense, $\tau$ controls the degree of consistency of $\hat{c}$ over the training steps. Higher $\tau$ ($\sim 1.0$) introduces relatively drastic changes of $\hat{c}$ during training, while smaller $\tau$ changes $\hat{c}$ conservatively. Fig. 8b illustrates the test performance of `EMA` with $\tau = \{1.0, 0.5, 0.1\}$ against `BPTO`. It is noteworthy that the forward processes of `EMA` with $\tau = 1.0$ and `BPTO` are identical, while the backward processes (i.e., the gradient calculation) are different. By comparing `EMA` ($\tau = 1.0$) and `BPTO`, we can observe that using a consistent $\hat{c}$ mechanism is crucial for successful training FOCA. Furthermore, as we decrease $\tau$ to $0.1$, FOCA exhibits better predictive performance.

## 6  CONCLUSION

Motivated by classical modeling-and-identification approaches, we introduced in this work a new contextual meta-learning method: FOCA. We separate the role of different parameters into a model, that captures context-invariant behavioral laws, and context, that specializes in finding instance-specific parameters. FOCA jointly learns model and context parameters by solving online bi-level optimization efficiently. Instead of relying on higher-order derivatives, it employs EMA for faster and more stable training. We empirically demonstrated in both static and dynamic regression tasks on complex dynamical systems that our approach outperforms or is competitive to baselines when evaluated on both in- and out-of-distribution contexts. We also performed ablation studies to examine the efficacy of our proposed training method.

FOCA is limited by modeling assumptions for the target system. Namely, FOCA seeks to model dynamical system families that share an underlying mathematical model $f_\theta$. If the target systems do not share consistent behavior, it would be hard to characterize a specific model family, and the performance of FOCA might degrade. Future work could develop a more general algorithm that allows for the adaptation of the function class at test time by changing $\theta$ when the common $f$ is not identifiable.

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

# A  DETAILS OF STATIC REGRESSION EXPERIMENTS

In this section, we provide the model architectures, training process, and additional experimental results for the polynomial regression task.

**Model architecture**  We employ `Encoder`, `CAVIA`, and `CoDA` as the baselines. All models use the same multi-layer perceptron (MLP) for $f_\theta$ and are trained with the same data batches for fair comparisons.

For brevity, we refer to a multi-layer perceptron (MLP) with hidden dimensions $n_1$, $n_2$, ... $n_l$ for each layer and hidden activation `act`, as `MLP(`$n_1$, $n_2$, ..., $n_l$`;` `act)`. We refer to a cross multi-head attention block (Vaswani et al., 2017) with $h$ heads and $x$ hidden dimensions as `X.MHA(`$h \times x$`)`.

Table A.1: Polynomial regression model architectures

|  | Context encoder / parameter generator | $f_\theta$ | Inner step $K$ | Inner step size $\alpha$ | $\tau$ |
|---|---|---|---|---|---|
| Encoder | MLP(2,32)/MLP(1,32)-X.MHA(4×32) | MLP(1+32, 64, 32, 1;LeakyReLU) | — | — | — |
| CAVIA | — | MLP(1+32, 64, 32, 1;LeakyReLU) | 5 | 1 | — |
| CoDA | $W : \mathbb{R}^{|c|} \to \mathbb{R}^{|\theta|}$ | MLP(1, 64, 32, 1;LeakyReLU) | 500 | 0.001 | — |
| FOCA | — | MLP(1+32, 64, 32, 1;LeakyReLU) | 100 | 0.001 | 0.1 |

Table A.1 summarizes the network architectures. The green colored values indicate the dimension of context $\hat{c}$. For `CAVIA`, we also perform hyperparameter search to optimize $K$. We found that `CAVIA` with $K > 5$ underperforms, as compared to $K = 5$. For `CoDA`, we set the dimension of context as 2 (i.e., $|c| = 2$), following the default setting of Kirchmeyer et al. (2022).

**Training details**  We train all models with mini-batches of 256 polynomials for 4,048 epochs using Adam (Kingma & Ba, 2015) with an initial learning rate of 0.001. The learning rate is scheduled by the cosine annealing method (Loshchilov & Hutter, 2017).

# B  DETAILS OF MASS SPRING (MS) EXPERIMENTS

In this section, we provide the model architectures, training process, and additional experimental results for the mass-spring systems.

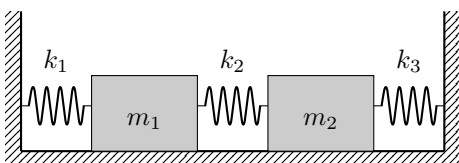

Figure A.1: Target mass-spring system. The models require to adapt to the change of spring constants $(k_1, k_2, k_3)$ and masses $(m_1, m_2)$.

We consider a frictionless three-spring two-mass system shown in Fig. A.1 with mass positions $x_1$ and $x_2$ governed by the following second-order ODE:

$$\begin{aligned}
\frac{\mathrm{d}^2 x_1}{\mathrm{d}t^2} &= -\frac{k_1 + k_2}{m_1} x_1 + \frac{k_2}{m_1} x_2 \\
\frac{\mathrm{d}^2 x_2}{\mathrm{d}t^2} &= -\frac{k_2 + k_3}{m_2} x_2 + \frac{k_2}{m_2} x_1
\end{aligned} \quad (A.1)$$

where $K$ is a coefficient matrix with spring constants $(k_1, k_2, k_3)$ and masses $(m_1, m_2)$.

**Data generation**  For training data generation, we generate 128 mass-spring systems whose parameters $(k_1, k_2, k_3, m_1, m_2)$ are sampled from $\mathcal{U}(0.75, 1.25)^5$ and numerically solve the mass-spring systems with Runge–Kutta 45 for $T = 10$ seconds with $\Delta t = 0.15$ second time intervals.

**Model architecture**   For $f_\theta$, we employ the 1D-CNN model from Brandstetter et al. (2021), which stacks an MLP, 1D-CNN, and consistency decoder (Brandstetter et al., 2021) with the bundling parameter $N = 25$. For brevity, we refer to a 1D-CNN layer with $x$ input channels, $y$ output channels, and filter size $w$ as $\texttt{Conv}(x, y, w)$, a consistency decoder as $\texttt{C.Dec}$, and the bi-directional GRU (Cho et al., 2014) with hidden dimension $x$ as $\texttt{GRU}(x)$.

Table A.2: Mass-spring prediction model architectures

| | Context encoder / parameter generator | $f_\theta$ |
|---|---|---|
| `Encoder` | GRU(64)-MLP(64, 64) | MLP(4×25+64, 64, 32, 4×25;LeakyReLU)-Conv(4, 4, 8)-LeakyReLU-Conv(4,4,1)-C.Dec |
| `CAVIA` | − | MLP(4×25+64, 64, 32, 4×25;LeakyReLU)-Conv(4, 4, 8)-LeakyReLU-Conv(4,4,1)-C.Dec |
| `CoDA` | $W : \mathbb{R}^{|c|} \to \mathbb{R}^{|\theta|}$ | MLP(4×25, 64, 32, 4×25;LeakyReLU)-Conv(4, 4, 8)-LeakyReLU-Conv(4,4,1)-C.Dec |
| `FOCA` | − | MLP(4×25+64, 64, 32, 4×25;LeakyReLU)-Conv(4, 4, 8)-LeakyReLU-Conv(4,4,1)-C.Dec |

Table A.2 summarizes the network architectures. The green colored values indicate the dimension of context $\hat{c}$. We set the inner step $K$ and step size $\alpha$ as 100/1 and 0.001/1 for `FOCA` and `CAVIA`, respectively, and $\tau$ as 0.1. For `CoDA`, we set the dimension of context as 2 (i.e., $|c| = 2$), following the default setting of Kirchmeyer et al. (2022).

**Training details**   We train all models with mini-batches of size 512 for 1,000 epochs using Adam (Kingma & Ba, 2015) with the initial learning rate of 0.001 and the pushforward regularization (Brandstetter et al., 2021). We use the past observations from the previous $2N$ to $N$ steps as the input of the adaptation process. For CoDA, we performed a hyperparameter search and found that $L1$ regularization on the norm of $c$ as 1e-3 and $W$ as 1e-7 performs well.

**Evaluation setting**   For in-training evaluations, we generate 2,048 mass-spring systems whose parameters $(k_1, k_2, k_3, m_1, m_2)$ are sampled from $\mathcal{U}(0.75, 1.25)^5$ and numerically solve the mass-spring system with Runge–Kutta 45 with $T = 5.0$ and $\Delta t = 0.01$. For out-of-training evaluations, we generate 2,048 mass-spring systems whose parameters $(k_1, k_2, k_3, m_1, m_2)$ are sampled from $\mathcal{U}(0.60, 0.75)^5 \cup \mathcal{U}(1.25, 1.35)^5$ for $T = 5.0$ with $\Delta t = 0.01$. All models take the first and second 0.25 seconds of observation for the task adaptation and predict 4.5 seconds of future states via the model rollout.

**Additional results**   We provide the visualization of the model generalization errors to the change of $m_1, m_2$ on the mass-spring systems, as shown in Fig. A.2.

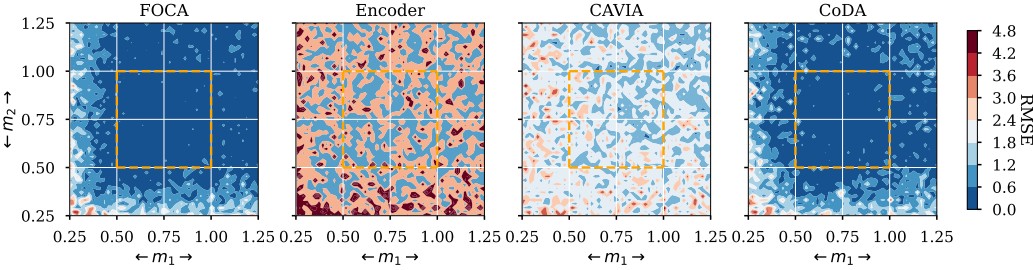

Figure A.2: In/out-of-distribution losses (in RMSE) on MS. Two parameters of MS $(m_1, m_2)$ are generated from $[0.25, 1.25]^2$ while the other two parameters are fixed to 0.75. The orange box indicates the boundaries of the training parameter distribution.

## C   DETAILS OF LOTKA-VOLTERRA (LV) EXPERIMENTS

In this section, we provide the model architectures, training process, and additional experimental results for the Lokta-Volterra systems (Lotka, 1910).

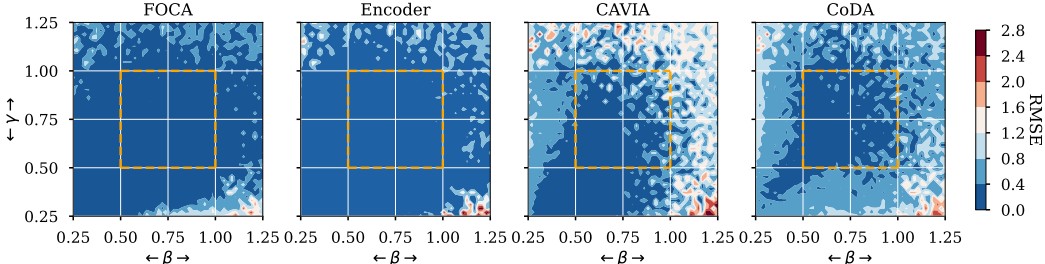

Figure A.3: In/out-of-distribution losses (in RMSE) on LV. Two parameters of LV $(\beta, \gamma)$ are generated from $[0.25, 1.25]^2$ while the other two parameters are fixed to 0.5. The orange box indicates the boundaries of the training parameter distribution.

The Lokta-Volterra system describes the interaction between a prey-predator pair in an ecosystem, formalized into the following ODE:

$$
\begin{aligned}
\frac{\mathrm{d}x}{\mathrm{d}t} &= \alpha x - \beta xy, \\
\frac{\mathrm{d}y}{\mathrm{d}t} &= \delta xy - \gamma y,
\end{aligned}
\tag{A.2}
$$

where $x, y$ are respectively the quantity of the prey and the predator and $\alpha, \beta, \gamma, \delta$ define how two species interact.

**Data generation**  For training data generation, we generate 128 Lokta-Volterra systems whose parameters $(\alpha, \delta) = (0.5, 0.5)$ and $(\beta, \gamma) \sim \mathcal{U}(0.5, 1.0)^2$ following Kirchmeyer et al. (2022). We then numerically solve the systems with Runge–Kutta 45 for $T = 50.0$ seconds with $\Delta t = 0.5$ second time intervals.

**Model architecture**  We employ the same network architectures of the MS experiments except with a different input dimensions of 2 and bundling parameter $N = 20$.

**Training details**  We train all models with mini-batches of size 128 for 5,000 epochs. Other details are kept the same as mass-spring experiments.

**Evaluation setting**  For in-training evaluations, we generate 2,048 Lokta-Volterra systems similarly to the training data generation. For out-of-training evaluations, we generate 2,048 Lokta-Volterra systems whose parameters $(\alpha, \delta) = (0.5, 0.5)$ and $(\beta, \gamma) \sim \mathcal{U}(0.25, 0.5)^2 \cup \mathcal{U}(1.0, 1.25)^2$ for $T = 50.0$ with $\Delta t = 0.5$. All models take the first and second 10.0 seconds of observation for the task adaptation and predict 30.0 seconds future states via the model rollout.

## D  DETAILS OF GLYCOLYTIC OSCILLATOR (GO) EXPERIMENTS

In this section, we provide the model architectures, training process, and additional experimental results for the glycolytic oscillators (Daniels & Nemenman, 2015).

The glycolytic oscillators describe yeast glycolysis dynamics with the following ODE:

$$
\begin{aligned}
\frac{\mathrm{d}S_1}{\mathrm{d}t} &= J_0 - \frac{k_1 S_1 S_6}{1 + (1/K_1^q)S_6^q} \\
\frac{\mathrm{d}S_2}{\mathrm{d}t} &= 2\frac{k_1 S_1 S_6}{1 + (1/K_1^q)S_6^q} - k_2 S_2(N - S_5) - k_6 S_2 S_5 \\
\frac{\mathrm{d}S_3}{\mathrm{d}t} &= k_2 S_2(N - S_5) - k_3 S_3(A - S_6) \\
\frac{\mathrm{d}S_4}{\mathrm{d}t} &= k_3 S_3(A - S_6) - k_4 S_4 S_5 - \kappa(S_4 - S_7) \\
\frac{\mathrm{d}S_5}{\mathrm{d}t} &= k_2 S_2(N - S_5) - k_4 S_4 S_5 - k_6 S_2 S_5 \\
\frac{\mathrm{d}S_6}{\mathrm{d}t} &= -2\frac{k_1 S_1 S_6}{1 + (1/K_1^q)S_6^q} + 2k_3 S_3(A - S_6) - k_5 S_6 \\
\frac{\mathrm{d}S_7}{\mathrm{d}t} &= \psi\kappa(S_4 - S_7) - kS_7,
\end{aligned}
\tag{A.3}
$$

where $S_1, S_2, S_3, S_4, S_5, S_6, S_7$ (states) represent the concentrations of 7 biochemical species and $J_0, k_1, k_2, k_3, k_4, k_5, k_6, K_1, q, N, A, \kappa, \psi$ and $k$ are the parameters of the glycolytic oscillators.

**Data generation**    For training data generation, we generate 128 glycolytic oscillators with the fixed parameters $J_0 = 2.5, k_2 = 6, k_3 = 16, k_4 = 100, k_5 = 1.28, k_6 = 12, q = 4, N = 1, A = 4, \kappa = 13, \psi = 0.1$ and $k = 1.8$ by sampling integer $k_1 \sim \mathcal{U}(80, 100)$ and $K_1 \sim \mathcal{U}(0.5, 1.0)$. We adopt the values or ranges of the parameters from Kirchmeyer et al. (2022). We then numerically solve the systems with Runge–Kutta 45 for $T = 5.0$ seconds with $\Delta t = 0.05$ second time intervals.

**Model architecture**    We employ the same network architectures of the MS experiments except with a different input dimensions of 7 and bundling parameter $N = 10$. For CoDA, we set the context dimension as 3.

**Training details**    We train all models with mini-batches of size 512 for 5,000 epochs. Other details are kept the same as mass-spring experiments.

**Evaluation setting**    For in-training evaluations, we generate 2,048 glycolytic oscillators similarly to the training data generation. For out-of-training evaluations, we generate 2,048 glycolytic oscillators whose parameters $k_1 \sim \mathcal{U}(75, 80) \cup \mathcal{U}(100, 105)$ and $K_1 \sim \mathcal{U}(0.45, 0.5) \cup \mathcal{U}(1.00, 1.05)$ and $(J_0, k_2, k_3, k_4, k_5, k_5, k_6, q, N, A, \kappa, \psi, k) = (2.5, 6, 16, 100, 1.28, 12, 4, 1, 4, 13, 0.1, 1.8)$ for $T = 5.0$ with $\Delta t = 0.05$. All models take the first and second 0.5 seconds of observation for the task adaptation and predict 4.5 seconds future states via the model rollout.

**Additional results**    We provide the visualization of the model generalization errors to the change of $k_1, K_1$ on the glycolytic oscillators, as shown in Fig. A.4.

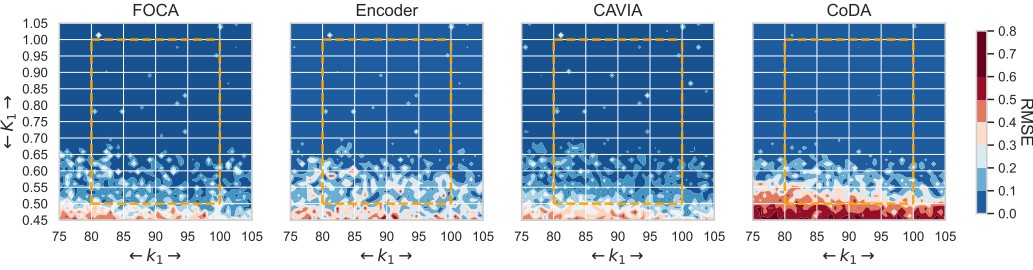

Figure A.4: In/out-of-distribution losses (in RMSE) on GO. The orange box indicates the boundaries of the training parameter distribution.

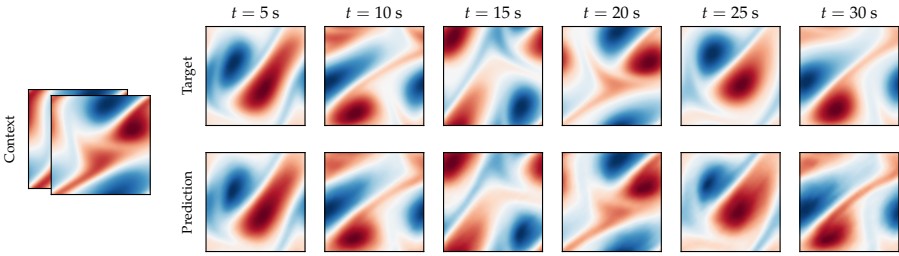

Figure A.5: Out-of-distribution generalization on the Navier-Stokes equations: FOCA is trained with viscosity $\nu \in [8 \times 10^{-4}, 1.2 \times 10^{-3}]$ and tested with $\nu = 6 \times 10^{-4}$. The model can reconstruct states even after long autoregressive rollouts in unseen conditions.

# E    DETAILS OF NAVIER-STOKES (NS) EXPERIMENTS

In this section, we provide the model architectures, training process, and additional experimental results for modeling the 2D Navier-Stokes equations (Stokes, 1851).

The Navier-Stokes equations describe the dynamics of incompressible flows with a two-dimensional PDE. In vorticity form they can be written as:

$$
\begin{aligned}
\frac{\partial w}{\partial t} &= -v\nabla w + \nu \Delta w + f \\
\nabla v &= 0 \\
w &= \nabla \times v
\end{aligned}
\tag{A.4}
$$

where $v$ is the velocity field and $w$ is the vorticity, $\nu$ is the viscosity, and $f$ is a forcing term. The domain is subject to periodic boundary conditions.

**Data generation**    We generate trajectories with a temporal resolution of $\Delta t = 1$ and a time horizon of $t = 10$. The space is discretized on a $32 \times 32$ grid and we set $f(x,y) = 0.1(\sin(2\pi(x + y)) + \cos(2\pi(x + y)))$, where $x, y$ are coordinates on the discretized domain (Yin et al., 2021). For training data, similarly to Kirchmeyer et al. (2022), we consider 5 training environments with $\nu \in \{8 \cdot 10^{-4}, 9 \cdot 10^{-4}, 1.0 \cdot 10^{-3}, 1.1 \cdot 10^{-3}, 1.2 \cdot 10^{-3}\}$ respectively. Each environment contains a total of 100 different initial conditions for a total of 500 training sequences.

**Model architecture**    For this experiment, we employ the Fourier Neural Operator (FNO) (Li et al., 2021). Thanks to their layers in the spectral domain and frequency mode pruning, frequency domain models have been proven to be well suited to model complex dynamical systems characterized by a multitude of natural frequencies (Pathak et al., 2022; Poli et al., 2022). We employ an FNO model with 4 spectral convolution layers, hidden layers with width 10, and frequency mode pruning set to the 12 highest frequencies. Temporal bundling is set to $N = 1$. We set the context dimension of CoDA to 3. For models whose input includes the context, i.e. Encoder, CAVIA and FOCA, we keep the same context dimension $|\hat{c}| = 64$ as in the previous experiments; the inferred context passes through a linear layer and resized to the grid size of $32 \times 32$ as an additional input channel of the FNO.

**Training details**    We train all models with mini-batches of size 16 for 100 epochs. Other details are kept the same as in the mass-spring experiments.

**Evaluation setting**    For in-training adaptation, we consider 4 environments with viscosity $\nu \in \{8.5 \cdot 10^{-4}, 9.5 \cdot 10^{-4}, 1.05 \cdot 10^{-3}, 1.15 \cdot 10^{-3}\}$ while for out-of-distribution adaptation we generate data from 4 environments with unseen ranges of viscosity, i.e. $\nu \in \{7.0 \cdot 10^{-4}, 7.5 \cdot 10^{-4}, 1.25 \cdot 10^{-3}, 1.30 \cdot 10^{-3}\}$. Each adaptation environment is generated out of 30 different initial conditions. All the models take as input the first two steps (i.e. the first 2 seconds) and predict the rest of the sequence with autoregressive rollouts.

**Additional results**   We provide in Fig. A.5 an additional qualitative visualization showing the out-of-distribution generalization on the Navier-Stokes equations. FOCA is trained with viscosity $\nu \in [8 \times 10^{-4}, 1.2 \times 10^{-3}]$ and tested with $\nu = 6 \times 10^{-4}$, demonstrating its adaptation capability in challenging and unseen settings.

