# OpenReview forum: "First-order Context-based Adaptation for Generalizing to New Dynamical Systems"
_ICLR.cc/2023/Conference — Submitted to ICLR 2023_

### Official Review · Reviewer_fp24 · 2022-10-25

**Confidence:** 4
**Clarity, Quality, Novelty And Reproducibility:** All good in this part.
**Correctness:** 4
**Technical Novelty And Significance:** 4
**Empirical Novelty And Significance:** 3
**Recommendation:** 8

**Strength And Weaknesses:**

Pros:
This paper is well-written and present a new contextual meta-learning method FOCA.  FOCA jointly learns model and context parameters by solving online bi-level optimization efficiently. Instead of relying on higher-order derivatives, it employs EMA for faster and more stable training. The authors empirically demonstrated in both static and dynamic regression tasks on complex dynamical systems that FOCA outperforms or is competitive to baselines when evaluated on both in- and out-of-distribution contexts.
Cons:
As the authors say in the last part,  FOCA only models systems that are identifiable from the given dataset. In the setting where the systems are sampled from nearly identical systems, FOCA may not be able to infer context properly.

**Summary Of The Paper:**

The authors  propose FOCA (First-Order Context-based Adaptation), a learning framework to model sets of systems governed by common but unknown laws that differentiate themselves by their context. FOCA  learns to represent the common law through shared parameters and relies on online optimization to compute system-specific context and its training involves a bi-level optimization problem. This paper use an exponential moving average (EMA)-based method that allows for fast training using only first-order derivatives.

**Summary Of The Review:**

I would suggest to accept this paper. It certainly is above the average.

---

> ### Author Response · Authors · 2022-11-13
> **Reply to reviewer fp24**
>
> We thank the reviewer for the feedback and for acknowledging the strengths of our work.
>
> ___
>
> We agree that the limitation mentioned in the conclusion is misleading. Therefore, we modify that part as follows:
>
> The limitation of our work comes from the assumption regarding the target systems that FOCA aims to model. Currently, FOCA seeks to model the dynamical systems where the system is characterized by a common mathematical model $f$ (parameterized by $\theta$) and a context $c$. If the target systems do not share consistent behavior, it is hard to be characterized by a specific form of an equation. As a result, the performance of FOCA might degrade. In future studies, we plan to develop a more general algorithm that allows the adaptation over the function classes by allowing it to change $\theta$ when the common $f$ is not identifiable.

---

### Official Review · Reviewer_dXrp · 2022-10-29

**Confidence:** 4
**Clarity, Quality, Novelty And Reproducibility:** The presentation of the manuscript is…
**Correctness:** 3
**Technical Novelty And Significance:** 2
**Empirical Novelty And Significance:** Not applicable
**Recommendation:** 5

**Strength And Weaknesses:**

The most confusing part of this manuscript is the *dynamical system* that is claimed to be adapting. But throughout the derivation I'm not seeing why it is even a dynamical system and what makes the adaptation of a dynamical system special in the derivation. I suspect that this is the largest weakness of the manuscript. Some detailed questions follow.

Questions:

1. As suggested by the title of this paper, FOCA should be a method that focuses on the generalization of new dynamical systems. The referred related works such as GBML and CoDA are also works that are specific to dynamical systems. However, from the problem setting described in the paper, it is not very clear to me how $f$ is a dynamic system (or perhaps the relationship between $x$ and $y$). This characterization will be very important as it will set apart this problem from the general setting of Meta-learning, where the datasets are often required to have i.i.d. data.
2. Following the last question, when $f$ represents a general dynamic system where data are not i.i.d. Solving (6) or line 9 of the algorithm with mean squared loss is perhaps non-trivial as one can no longer directly uses ERM which assumes data are i.i.d. It is indeed discussed in the CoDA paper that in general the MSE objective of (6) is intractable and can only be approximated.
3. In section 3, it is stated that the learning objective for meta-learning is to “minimize the generalization error on $D_{tr}$. This is confusing as usually the generalization error is defined with the testing dataset while $D_{tr}$ denotes the training dataset. The paper actually did not define the test dataset for the generalization problem studied, thus this left me confused.
4. For the time-series prediction tasks, FOCA is compared against CoDA, CAVIA, and encoder. However, it seems that CAVIA and encoder were originally designed for the general meta-learning setting and thus it is reasonable to have suboptimal performance for generalization to new dynamical systems. Therefore FOCA can be evaluated more comprehensively by comparing it with other algorithms designed specifically for learning dynamical systems, such as [1] and [2]. Besides, in the CoDA paper, two versions of the algorithm are proposed (with $\ell_1$ or $\ell_2$ constraints), could you state which version of the CoDA is used in this work as a baseline?

[1]: Yin, Y., Ayed, I., de Bezenac, E., Baskiotis, N., and Gallinari, P. LEADS: Learning dynamical systems that generalize across environments. In Beygelzimer, A., Dauphin, Y., Liang, P., and Vaughan, J. W. Advances in Neural Information Processing Systems, 2021a
[2]: Wang, R., Walters, R., and Yu, R. Meta-learning dynamics forecasting using task inference. CoRR, abs/2102.10271, 2021c.

Miscellaneous questions:
1. For Table 1, it is better to include references to each algorithm in the caption. Otherwise, it can be a bit confusing (e.g. encoder can be a general terminology used while here it is used to refer to a specific algorithm?) The use of absolute value ($| \cdot |$) to donate memory usage is also confusing. To my understanding, $\theta$, $\psi$, and $W$ are parameters that can be a vector/matrix. Therefore it is unclear what the definition of the absolute value of a vector/matrix is.
2. For Figure 5, could you report the standard deviation of the prediction outcome obtained with the algorithms?



**Summary Of The Paper:**

The paper investigates the problem of generalizing to new dynamical systems through an efficient first-order context-based algorithm (FOCA). The algorithm is based on a bi-level learning objective obtained through the modeling-then-identification approach. An EMA step is taken to accelerate the training process.

**Summary Of The Review:**

The manuscript is overall in good quality but I'm not aware of its relevance towards dynamical systems and its novelty thereafter.

---

> ### Author Response · Authors · 2022-11-13
> **Reply to reviewer dXrp**
>
> Thank you for your review. Your input helps us to improve our manuscript. Here we provide our replies to your questions.
>
> ___
> **Clarifications about target systems $f$**
>
> Our notation $y=f_{\theta}(x,c)$ denotes a parameterized function mapping the input $x$ and context $c$ to the output $y$, where parameter $\theta$ represents a model class.
>
> In a typical regression setting, $x$ can be just an input value, and $y$ can be the output corresponding to input $x$ given context $c$. If we set $x=x_t$, and $y=x_{t+1}$, then $f$ can express how the future state $x_{t+1}$ is updated given the current state $x_t$ and the context $c$.
>
> ___
> **About the i.i.d assumption**
>
> As the reviewer suggested, clarifying our assumptions about our target dynamic system is beneficial. We assume that our target dynamic system is stationary given a specific context $c^i$ representing the parameters specifying dynamic system $i$ (i.e., task $i$). Thus, we do not assume that the context varies in the observed (sub) trajectory. We assume that trajectory data $\\{(x_n^i, y_n^i)\\}$ are independent for $i \neq j$ (i.e., task data set are independent). This independence assumption is used in optimizing the shared parameter $\theta$.
>
> In solving (6), we assume that each data point $\\{(x_n^i, y_n^i)\\}\in D^i$ are i.i.d given the context $c^i$. In other words, we assume that i.i.d noise $e_n$ for observed data, i.e., $y_n=f_{\theta}(x_n,c)+e_n$. If we assume $e_n$ is zero-mean Gaussian noise, Eq.(6) becomes a regression problem with MSE loss.
>
> ___
> **About $D_{\text{tr}}$ and the generalization error**
>
> Agreeing with the confusion, we updated $D_{\text{tr}}$ to $D_{\text{ev}}$ in our manuscript so that the partitioning is clearer over inference and evaluation for a single task. As pointed out, since we do not define the (meta)-test dataset, it is unsuitable to explain the loss in equation (3) as the generalization error, and we update the manuscript so that it only states “error on $D_{\text{ev}}$”.
>
> ___
> **Comparison with the additional baselines**
>
> LEADS [1] and DyAD [2] are meta-learning approaches for generalizing to new dynamical systems. We would like to mention that we consider the baselines that are not architecture-dependent for performance comparison; as mentioned in the paper, all baselines and FOCA utilize the same prediction model architecture for a fair comparison.
>
> The adaptation mechanism of DyAD [2] is (1) AdaIN, which adaptively changes the normalization factors of the normalization layers, and (2) AdaPad, which adaptively changes the padding values of convolution layers. Hence, to apply DyAD[2], the network must have normalization and convolutional layers. On the other hand, FOCA and baseline methods are applicable for general neural network architectures. Because of this, a direct comparison between DyAD and FOCA may not be possible. Unlike DyAD, LEADS [1] is applicable to various network architectures. The following table summarizes the LEADS results on the MS, LV, and GO datasets.
> |             |         MS        |                   |         LV        |                   |         GO        |                   |         NS        |                   |
> |:-----------:|:-----------------:|-------------------|:-----------------:|-------------------|:-----------------:|-------------------|:-----------------:|-------------------|
> |             |    In-training    | Out-of-training   | In-tranining      | Out-of-training   | In-training       | Out-of-training   | In-training       | Out-of-training   |
> |    LEADS    | $1.327 \pm 0.666$ | $1.633 \pm 0.863$ | $0.262 \pm 0.239$ | $0.812 \pm 0.990$ | $1.305 \pm 0.913$ | $1.355 \pm 1.072$ | $0.793 \pm 0.019$ | $0.792 \pm 0.011$ |
> | FOCA (ours) | $0.258 \pm 0.156$ | $0.478 \pm 0.295$ | $0.079 \pm 0.061$ | $0.303 \pm 0.291$ | $0.043 \pm 0.066$ | $0.147 \pm 0.168$ | $0.042 \pm 0.026$ | $0.070 \pm 0.038$ |
>
> [1] LEADS: Learning Dynamical Systems that Generalize Across Environments
>
> [2] Meta-Learning Dynamics Forecasting Using Task Inference
> ___
> **About CoDA hyperparameter**
>
> We tested CoDA models with/without $l_1$ and $l_2$ penalties. Similar to the CoDA paper, we confirmed CoDA with $l_1$ tends to perform better than the version with $l_2$. Hence, we report the performance of CoDA with $l_1$. We updated the appendix of the manuscript, which explains the training details of FOCA and baselines so that it explains the regularization of CoDA.
>
> ___
> **About the meaning of $|\cdot|$**
>
> We used $|\cdot|$ to denote the number of elements (cardinality) of $\cdot$. As you pointed out, $\cdot$ can generally be a tensor. We updated the manuscript so that $|\cdot|$ explicitly denotes the cardinality of $\cdot$.
>
> ___
> **About the standard deviation of prediction outcomes in Figure 5**
>
> Figure 5 describes one prediction example of a sampled MS system. We report the mean and standard deviation of the prediction errors obtained from 2048 test cases of MS systems for all algorithms in Table 2.

---

> > ### Author Response · Authors · 2022-12-08
> > **Request for feedback**
> >
> > As the author-reviewer discussion phase is coming to an end, please let us know if there are further questions or concerns to address. We will be happy to provide additional clarifications.

---

### Official Review · Reviewer_t6gz · 2022-11-08

**Confidence:** 3
**Correctness:** 2
**Technical Novelty And Significance:** 3
**Empirical Novelty And Significance:** 3
**Recommendation:** 5

**Clarity, Quality, Novelty And Reproducibility:**

The paper often seems to lack clarity in terms of the experimental setup and discussion of results. A few comments for example:
1. The axes in most plots are not labels or described which makes it difficult to judge the actual evaluation criteria.
2. In the static regression example, it is not clear as to what components $\theta$ and $c$ are capturing. Given the dimensionality of the c parameters $64$, it is not even clear if this is just a spurious overparameterization of the network as the range of linear or quadratic functions in Fig 4 doesn't seem to be diverse at all for a given $\theta$.
3. Distinguishing context based adaptation and a few-shot learning based meta-learning objective is important and hasn't been discussed well in the current draft. In the test phase, $\theta$ is kept fixed and only context inference is allowed. However, can the comparison with other procedures be made when a few steps for updating $\theta$ are also allowed?

**Strength And Weaknesses:**

Strengths:
1. The overall idea behind FOCA is well-motivated along with its connections of using a target network in RL and other works in supervised learning. Addressing the stability of bilevel optimization problems, even outside of meta-learning is a common issue, and this work provides empirical evidence of the efficacy of a simple method.
2. The empirical results for FOCA look quite promising on the datasets considered in the paper. However, there are related issues for comparison (see below).

Weaknesses:
1. Overall, despite the empirical focus of this paper, the discussion about weaknesses of other approaches to handle the dependence of best fit $\hat c^i$ is rather weak. Backpropagating through the optimization trajectory in MAML style methods is an issue but various alternatives have been studied in recent years, some of which are even referenced in this paper. Did the authors investigate any such variations empirically on these datasets? Further, implicit gradients 'ideally' would require solving the inner optimization accurately but can be used as an approximate way of solving the $\hat c$ dependency issue. However, I do agree that adding an exponentially moving average with large values of $\tau$ in this case would benefit as well. Also, can a simpler two timescale learning rate scheme for updating the inner variable $c$ at a slower rate work? It will be worth investigating that.
2. In the experiments, it is not clear as to what extent the context variables capture the task specific parameters vs shared parameters capturing the underlying functional form. For instance, is there any result on how the dimensionality of the $c$ affect performance when it is known that the actual cardinality of the context set if small. Similarly, changes in the overall architecture and investigating the distribution of errors across the tasks in the test set will indicate any undesired behavior.
3. Evaluating CAVIA: How is CAVIA run during test inference? Have the authors ran a few gradient steps $K$ during test time as well? If yes, can the alternative of finding the best fit $\hat c^i$ be tested as well? That way, we can actually decouple the effect of meta-training procedure and not induce indirect effects of the meta-test step.
4. Architectural differences with CODA: In the experiments, the authors use $\theta^i = \theta + W\hat c^i$ as the task-adapted parameter for CODA. This is starkly different from how the counterpart for FOCA is structured: $\hat f^i(x) = f_\theta(x,\hat c^i)$. The main difference between CODA and FOCA can be said to be joint training of $\theta,c$ vs interleaved training using EMA respectively. As such, I do not see the linear approximation for the model as anything inherent to CODA. I recommend the authors to have the same setup for comparison.


**Summary Of The Paper:**

The paper investigates a first order optimization-based framework for context-based meta-learning of static or dynamical systems. In the considered problem setup, the authors look into settings where a large class of models can be expressed as a combination of a function parametrized by shared parameters $\theta$ and task-specific context parameters $c$. The authors propose FOCA as the framework which uses typical first order optimization methods for online optimization for training with a key ingredient being the use of asymmetric configurations of $\theta$ to learn the shared parameters and the setting $\bar\theta$ used to identify the best context parameter set $c^i$ for any task $i$ during training. To maintain stability over training, the authors propose to use a slowly exponentially moving average of $\theta$ as a proxy for $\bar\theta$ instead of drastically moving best values of $\theta$. Finally, the authors test their proposed framework on a toy static regression problem and dynamical system datasets based on well-known ODE/PDE systems.

**Summary Of The Review:**

Overall, the paper investigates an interesting idea and shows promising empirical results. However, I feel that the paper needs improvement on the thoroughness of experiments, adequate comparison with potential baselines and a better ablation study of the key component providing the improvement (if stability is the main issue, it can be combined with other baselines as well). Further, the tests for whether the context captures task specific information is rather weak and needs further evidence.

NOTE: I want to take this space to note that the review is being submitted outside the stipulated timeframe, and I apologize from my end for the delay. I hope that the authors do find my comments helpful.

---

> ### Author Response · Authors · 2022-11-13
> **Reply to reviewer t6gz**
>
> Thank you for your review. Your input helps us to improve our manuscript. Here we provide our replies to your questions.
>
> ___
>
> **About the alternative approaches to MAML**
>
> The well-known limitations of MAML are (1) demanding computational cost originating from the higher-order gradient computation during (meta)-training, (2) meta-overfitting phenomena that are presumed to originate from the adaptation procedure of MAML which updates all learnable parameters $\theta$.
>
> We can think of approaches that aim to overcome the computational limitations of MAML by (1) updating the subset of meta (network) parameter $\theta$ [1], (2) training a model to generate updated parameters [2], (3) applying some approximations (most likely first-order methods) [3], or (4) applying implicit differentiation to compute the (meta) gradient [4, 5].
>
> In the manuscript, we compare FOCA with the methods that are the instances of the listed approaches.
>
> - CAVIA is an example of (1) as it finds the small-sized $c$ via MAML-like updates.
> - CoDA is an example of (2) as it trains the linear basis $W$so that later it uses $W$ to generate updated parameters.
> - FO — one of the training ablation models in Figure 8 — is an example of (3) as it approximates the higher-order gradient computation via a first-order method.
> - Implicit — one of the training ablation models in Figure 8 — is an example of (4) as it calculates the (meta) gradient via the implicit differentiation.
>
> To overcome the second limitation, various methods apply the idea of providing small dimensional vectors (e.g., the subset of parameters or additional context input) and empirically show better meta-test performances. Similar to the aforementioned approaches, FOCA utilizes the context $c$, a small dimensional vector, to adapt to new systems. FOCA finds the (most) appropriate $c$ by solving its inner optimization problem.
>
> [1] Rapid Learning or Feature Reuse? Towards Understanding the Effectiveness of MAML
>
> [2] Meta-Learning with Latent Embedding Optimization
>
> [3] On First-Order Meta-Learning Algorithms
>
> [4] Meta-Learning with Implicit Gradients
>
> [5] Continuous-Time Meta-Learning with Forward Mode Differentiation
> ___
>
> **About the two-time scale learning rate scheme**
>
>
> We comprehend the mentioned “two-time scale learning rate scheme” as the collection of the methods that use the different learning rates for training two models (e.g., Generator and discriminator training of GAN, Critic and actor training of reinforcement learning, or Student and teacher network training of (self/semi) supervised learning)
>
> FOCA only trains $f_\theta$ and the context $c$ is found by solving equation 6, and we don’t store or train $\hat c^i$. The context finding is done online when $f_\theta$ adapts to the new systems. If we misunderstood your comments, can you clarify the “two-time scale learning rate scheme” in our problem setting?

---

> > ### Author Response · Authors · 2022-11-13
> > **Reply to reviewer t6gz [2]**
> >
> > **About the effect of $c$ dimensions**
> >
> > We conduct additional experiments that show how the dimension of $c$ affects the prediction performance. The following tables summarize the experimental results on the MS and LV datasets.
> >
> > Table: the effect of context dimension in the mass-spring dataset
> >
> > | dim($c$) | In-training dist. | Out-of-training dist. |
> > |:--------:|:-----------------:|:---------------------:|
> > |     1    | $0.349 \pm 0.207$ | $0.758 \pm 0.505$     |
> > |     2    | $0.374 \pm 0.223$ | $0.772 \pm 0.471$     |
> > |     4    | $0.282 \pm 0.173$ | $0.565 \pm 0.375$     |
> > |     8    | $0.256 \pm 0.164$ | $0.522 \pm 0.356$     |
> > |    16    | $0.241 \pm 0.157$ | $0.462 \pm 0.310$     |
> > |    32    | $0.231 \pm 0.146$ | $0.480 \pm 0.327$     |
> > |    64    | $0.258 \pm 0.156$ | $0.478 \pm 0.295$     |
> >
> > In the MS dataset, the model with $dim(c) \geq 5$ shows better predictive performance than the models with $\text{dim}(c) <4$. Note that the number of free parameters of MS is 5.
> >
> > Table: the effect of context dimension in the Lotka-Volterra dataset
> >
> > | dim($c$) | In-training dist. | Out-of-training dist. |
> > |:--------:|:-----------------:|:---------------------:|
> > |     1    | $0.078 \pm 0.062$ | $0.390 \pm 0.390$     |
> > |     2    | $0.070 \pm 0.057$ | $0.380 \pm 0.362$     |
> > |     4    | $0.091 \pm 0.087$ | $0.426 \pm 0.434$     |
> > |     8    | $0.078 \pm 0.051$ | $0.363 \pm 0.361$     |
> > |    16    | $0.070 \pm 0.055$ | $0.398 \pm 0.385$     |
> > |    32    | $0.069 \pm 0.057$ | $0.415 \pm 0.426$     |
> > |    64    | $0.079 \pm 0.061$ | $0.303 \pm 0.291$     |
> >
> > In the LV dataset, the model with $\text{dim}(c) \geq 2$ shows better predictive performance than the models with $\text{dim}(c) < 2$. Note that the number of free parameters of LV is 2. (i.e., $\alpha$ and $\gamma$)
> >
> > ___
> > **About CAVIA evaluation**
> >
> > We run $K$ times the gradient steps for CAVIA adaptation. As far as we know, for evaluating CAVIA, taking as many $K$ gradient steps as the meta-training steps is standard, as also done in the [official implementation](https://github.com/lmzintgraf/cavia/blob/91f093af9d6f463ee651db533f6c2acc637c7e9f/regression/cavia.py#L179).
> >
> > We evaluate CAVIA and FOCA that take $K_\text{train}$ gradient steps during meta-training and use $K_\text{test}$ gradient steps for meta-testing. The following table summarizes the results.
> >
> > | CAVIA | $K_{\text{test}}=1$ | $K_{\text{test}}=5$ | $K_{\text{test}}=10$ | $K_{\text{test}}=50$ |
> > |:---:|:---:|:---:|:---:|:---:|
> > | $K_{\text{train}}=1$ | $0.347 \pm 0.249$ | $0.325 \pm 0.224$ | $0.300 \pm 0.194$ | $0.228 \pm 0.101$ |
> > | $K_{\text{train}}=5$ | $0.363 \pm 0.263$ | $0.348 \pm 0.251$ | $0.328 \pm 0.234$ | $0.219 \pm 0.133$ |
> > | $K_{\text{train}}=10$ | $0.409 \pm 0.274$ | $0.401 \pm 0.268$ | $0.388 \pm 0.257$ | $0.276 \pm 0.148$ |
> > | $K_{\text{train}}=50$ | $0.493 \pm 0.218$ | $0.467 \pm 0.209$ | $0.437 \pm 0.197$ | $0.271 \pm 0.135$ |
> >
> > | FOCA | $K_{\text{test}}=1$ | $K_{\text{test}}=5$ | $K_{\text{test}}=10$ | $K_{\text{test}}=50$ |
> > |:---:|:---:|:---:|:---:|:---:|
> > | $K_{\text{train}}=1$ | $0.059 \pm 0.047$ | $0.064 \pm 0.050$ | $0.068 \pm 0.056$ | $0.077 \pm 0.075$ |
> > | $K_{\text{train}}=5$ | $0.071 \pm 0.074$ | $0.071 \pm 0.077$ | $0.073 \pm 0.082$ | $0.079 \pm 0.097$ |
> > | $K_{\text{train}}=10$ | $0.069 \pm 0.061$ | $0.068 \pm 0.058$ | $0.068 \pm 0.056$ | $0.071 \pm 0.065$ |
> > | $K_{\text{train}}=50$ | $0.062 \pm 0.056$ | $0.061 \pm 0.054$ | $0.061 \pm 0.054$ | $0.061 \pm 0.055$ |
> >
> > As we can confirm from the table, none of the CAVIA models outperforms FOCA even if they use larger $K_\text{train}$ and/or $K_\text{test}$.
> >
> > ___
> > **About the baseline selections**
> >
> > We choose the baselines (i.e., Encoder, CAVIA, CoDA) as their adaptation mechanism is to find the lower-dimensional context $c$ rather than **directly** changing the entire model parameters $\theta$.
> >
> > We agree that the adaptation of CoDA involves the change of the entire $\theta$. However, once trained, its adaptation only depends on the low dimensional $c$.
> >
> > Furthermore, as mentioned in the manuscript, we chose CoDA as a baseline since CODA and FOCA aim to tackle similar problems of designing a context-based model generalizable to different physical systems sharing structural similarity. In that sense, CoDA is one of the most comparable methods. CAVIA has a similar structure to FOCA but uses a different training scheme.
> >
> > ___
> > **About the figure labels**
> >
> > Thank you for the feedback. We have updated the figures with the $x$ and $y$-axis labels.

---

> > > ### Author Response · Authors · 2022-11-13
> > > **Reply to reviewer t6gz [3]**
> > >
> > > **About the role of $c$ and $\theta$**
> > >
> > > We hypothesize the shared parameters $\theta$ capture the underlying functional form, and the context variables $c$ capture the task-specific parameters. We verify this hypothesis on a simple polynomial regression fit.
> > >
> > > Figure 4 shows that $f_{\theta}(\cdot, c)$, with the given $\theta$, models the functions of similar class, and $f_{\theta}(\cdot, c)$ models the specific function of the class by providing the corresponding $c$ to it. For instance, $f_{\theta(0)}(\cdot, \cdot)$ and $f_{\theta(1)}(\cdot, \cdot)$ models linear functions and quadratic functions, respectively, and $c$ controls the slope and curvature of linear functions and quadratic functions, respectively.
> > >
> > > To verify the role of $\theta$ more clearly, we have conducted an additional experiment. We hypothesize that if the meta parameter $\theta$  is fixed, the function $f_{\theta}(\cdot, c)$ will not fit well to the data set generated from the different function class, regardless of $c$.
> > >
> > > Let us denote $f_{\theta(0)}(\cdot, c)$ the function trained from the data sampled from the linear function $y=ax$ with  $0.1 \leq a \leq 1.5$
> > > and $f_{\theta(1)}(\cdot, c)$ the function trained from the data sampled from the quadratic function $y=ax^2$ with  $0.1 \leq a \leq 1.5$
> > >
> > > To verify our hypothesis, we optimize the task-specific parameter $c$ of $f_{\theta(0)}(\cdot, c)$, whose meta parameter is fixed as $\theta(0)$ for linear, with the data sampled from the quadratic function. The figure in **[the link](http://bit.ly/3ExaRMm)** shows that the fitted functions are still linear; optimizing $c$ does not change the function class.
> > >
> > > Similarly, we optimize the task-specific parameter $c$ of $f_{\theta(1)}(\cdot, c)$, whose meta parameter is fixed as $\theta(1)$ for quadratic, with the data sampled from the linear function. The figure in **[the link](http://bit.ly/3Ea7utt)** shows that the fitted functions are still quadratic; optimizing $c$ does not change the function class.
> > >
> > > The results support the roles of $\theta$ and $c$ in FOCA; $\theta$ capture the function class information and $\theta$ dedicates to the specific function in the class.
> > >
> > > Additionally, we prepare videos that visualize how the change of $c$ affects the state trajectory of the mass-spring system and the Navier-Stokes equation. The first video shows that the change of $c$ founds the target systems while the overall curve forms are maintained (i.e., the effect of fixed $\theta$). We visualize the first two dimension of $c$ ($c_0$ and $c_1$) and don't apply embedding methods (e.g., PCA) on $c$ as the result can change to the selection of embedding vectors. The second video visualizes a similar process on an NS system.
> > >
> > > **[Mass-spring video](https://tinyurl.com/yw6x3rky)**
> > >
> > > **[Navier-Stokes video](https://tinyurl.com/ymrc2mju)**
> > >
> > > ___
> > > **Discussion about context-based adaptation and few-shot learning-based meta-learning**
> > >
> > > We can classify meta-learning approaches into (1) the approaches that update all of $\theta$ (i.e., few-shot learning-based meta-learning)  and (2) the approaches that update (or find) the small dimensional vector $c$ depending (i.e., context-based adaptation) on their adaptation mechanism.
> > >
> > > In the manuscript, we mainly compare FOCA with the approaches of the second type. The encoder approach utilizes an encoder network to infer $c$. CAVIA takes a few steps of the gradient updates to infer $c$. CoDA solves its adaptation problem to infer $c$.
> > >
> > > As the reviewer mentioned, we can compare FOCA with the first types that allow updating all $\theta$ during adaptation. We employ MAML [1] and LEADS [2] as the baselines.
> > >
> > > |             |         MS        |                   |         LV        |                   |         GO        |                   |         NS        |                   |
> > > |:-----------:|:-----------------:|-------------------|:-----------------:|-------------------|:-----------------:|-------------------|:-----------------:|-------------------|
> > > |             |    In-training    | Out-of-training   | In-tranining      | Out-of-training   | In-training       | Out-of-training   | In-training       | Out-of-training   |
> > > |     MAML    | $6.628 \pm 2.974$ | $6.825 \pm 8.781$ | $0.075 \pm 0.074$ | $0.456 \pm 0.443$ | $0.049 \pm 0.065$ | $0.189 \pm 0.180$ | $0.099 \pm 0.065$ | $0.122 \pm 0.180$ |
> > > |    LEADS    | $1.327 \pm 0.666$ | $1.633 \pm 0.863$ | $0.262 \pm 0.239$ | $0.812 \pm 0.990$ | $1.305 \pm 0.913$ | $1.355 \pm 1.072$ | $0.793 \pm 0.019$ | $0.792 \pm 0.011$ |
> > > | FOCA (ours) | $0.258 \pm 0.156$ | $0.478 \pm 0.295$ | $0.079 \pm 0.061$ | $0.303 \pm 0.291$ | $0.043 \pm 0.066$ | $0.147 \pm 0.168$ | $0.042 \pm 0.026$ | $0.070 \pm 0.038$ |
> > >
> > > As shown in the table, the approaches only update $c$ (i.e., FOCA) generally show better prediction results. By comparing the result, we can verify the effectiveness of the adaptation mechanism that only updates $c$ when generalization targets share functional forms.

---

> > > > ### Author Response · Authors · 2022-12-08
> > > > **Request for feedback**
> > > >
> > > > As the author-reviewer discussion phase is coming to an end, please let us know if there are further questions or concerns to address. We will be happy to provide additional clarifications.

---

### Decision · Program_Chairs · 2023-01-20

**Decision:**

Reject

**Justification For Why Not Higher Score:**

* The paper advertises the technique for dynamical systems in general, but the formulation suggests that it is for static supervised learning.  The authors clarified in the rebuttal that by setting $y$ to $x_{t+1}$, we can model a dynamical system.  This is neat and correct.  However, this needs to be explained in the paper.  Also, the class of dynamical systems that can be modeled this way needs to be specified.  For instance it is not clear how to apply the approach to dynamical systems where there is an $x_t$ and $y_t$ at every time step.  Also, it is not clear whether the time between different time steps can vary.
* It is not always clear how the poposed technique compares to existing techniques.  For example, the empirical comparison wih CoDA uses a different neural architecture so it is not clear whether the better results are due to different neural architectures or the exponential moving average.  An apple-to-apple comparison is needed.  More generally, the take away of the experiments is not always clear.

Since those concerns are not critical, the paper could be accepted for publication if there is a need for more papers to be published.

**Justification For Why Not Lower Score:**

NA

**Metareview: Summary, Strengths And Weaknesses:**

The paper describes a new meta learning technique that avoids second order derivatives by using an exponential moving average trick.  This is quite interesting.

Strengths:
* The exponential moving average trick is novel and interesting
* The approach avoids second order derivatives while being is farily general and simple

Weaknesses:
* The paper advertises the technique for dynamical systems in general, but the formulation suggests that it is for static supervised learning.  The authors clarified in the rebuttal that by setting $y$ to $x_{t+1}$, we can model a dynamical system.  This is neat and correct.  However, this needs to be explained in the paper.  Also, the class of dynamical systems that can be modeled this way needs to be specified.  For instance it is not clear how to apply the approach to dynamical systems where there is an $x_t$ and $y_t$ at every time step.  Also, it is not clear whether the time between different time steps can vary.
* It is not always clear how the poposed technique compares to existing techniques.  For example, the empirical comparison wih CoDA uses a different neural architecture so it is not clear whether the better results are due to different neural architectures or the exponential moving average.  An apple-to-apple comparison is needed.  More generally, the take away of the experiments is not always clear.

Overall, this is very interesting and promising work.  If the paper is not accepted the authors are encouraged to continue this work by addressing the above points while taking into account the feedback of the reviewers.


**Summary Of Ac-Reviewer Meeting:**

The discussion focused on the following two concerns raised by two reviewers:

* The paper advertises the technique for dynamical systems in general, but the formulation suggests that it is for static supervised learning.  The authors clarified in the rebuttal that by setting $y$ to $x_{t+1}$, we can model a dynamical system.  This is neat and correct.  However, this needs to be explained in the paper.  Also, the class of dynamical systems that can be modeled this way needs to be specified.  For instance it is not clear how to apply the approach to dynamical systems where there is an $x_t$ and $y_t$ at every time step.  Also, it is not clear whether the time between different time steps can vary.
* It is not always clear how the poposed technique compares to existing techniques.  For example, the empirical comparison wih CoDA uses a different neural architecture so it is not clear whether the better results are due to different neural architectures or the exponential moving average.  An apple-to-apple comparison is needed.  More generally, the take away of the experiments is not always clear.

While these concerns are not critical, two of the reviewers felt that the authors should really address those concerns since they were raised in their initial review, but they were ignored by the authors in the rebuttal.  The third reviewer did not feel that these concerns were significant, but did not object to the rejection of the paper.  The consensus is to recommend weak rejection, while making it clear that the paper could be accepted if there is a need for more papers to be presented at the conference.